# Statistical Analysis of the Main Risk Factors of an Unfavorable Evolution in Gastroschisis

**DOI:** 10.3390/jpm11111168

**Published:** 2021-11-09

**Authors:** Elena Ţarcă, Solange Tamara Roșu, Elena Cojocaru, Laura Trandafir, Alina Costina Luca, Valeriu Vasile Lupu, Ștefana Maria Moisă, Valentin Munteanu, Lăcrămioara Ionela Butnariu, Viorel Ţarcă

**Affiliations:** 1Department of Surgery II—Pediatric Surgery, “Grigore T. Popa” University of Medicine and Pharmacy, 700115 Iaşi, Romania; elena.tuluc@umfiasi.ro; 2Department of Nursing, “Grigore T. Popa” University of Medicine and Pharmacy, 700115 Iaşi, Romania; solange.rosu@umfiasi.ro; 3Department of Morphofunctional Sciences I—Pathology, “Grigore T. Popa” University of Medicine and Pharmacy, 700115 Iaşi, Romania; elena2.cojocaru@umfiasi.ro; 4Department of Mother and Child Medicine—Pediatrics, “Grigore T. Popa” University of Medicine and Pharmacy, 700115 Iaşi, Romania; alina.luca@umfiasi.ro (A.C.L.); stefana-maria.moisa@umfiasi.ro (Ș.M.M.); 5Department of Biomedical Sciences, “Grigore T. Popa” University of Medicine and Pharmacy, 700115 Iaşi, Romania; valentin_munteanu2002@yahoo.com; 6Department of Mother and Child Medicine—Genetics, “Grigore T. Popa” University of Medicine and Pharmacy, 700115 Iaşi, Romania; ionela.butnariu@umfiasi.ro; 7County Statistics Department, 700115 Iaşi, Romania; vtarca@gmail.com

**Keywords:** gastroschisis, morbidity, mortality, outcome

## Abstract

Gastroschisis is a congenital abdominal wall defect that presents an increasing occurrence at great cost for the health system. The aim of the study is to detect the main factors of an unfavorable evolution in the case of gastroschisis and to find the best predictors of death. Methods: we conducted a retrospective cohort study of neonates with gastroschisis treated in a tertiary pediatric center during the last 30 years; 159 patients were eligible for the study. Logistic regression was used to determine the risk of death, estimated based on independent variables previously validated by the chi-square test. Results: if the birth weight is below normal, then we find an increased risk (4.908 times) of evolution to death. Similarly, the risk of death is 7.782 times higher in the case of developing abdominal compartment syndrome, about 3 times in the case of sepsis and 7.883 times in the case of bronchopneumonia. All four independent variables contributed 47.6% to the risk of death. Conclusion: although in the past 30 years in our country we have seen transformational improvements in outcome of gastroschisis, survival rates increasing from 26% to 52%, some factors may still be ameliorated for a better outcome.

## 1. Introduction

Gastroschisis is a relatively benign malformation, but is associated with substantial perinatal morbidity and there is still no consensus about the best mode and time of delivery, about the algorithm for its treatment, and about the main risk factors for morbidities [1]. Although there has been a trend towards increasing occurrence of gastroschisis worldwide over the past 20–30 years, the relatively low overall prevalence (2–4 cases per 10,000 livebirths) makes it challenging to assemble large cohorts at individual centers in order to evaluate the associations between some independent variables and health outcomes of affected neonates [1,2]. The main techniques currently used for treatment are the primary closure of the defect with complete reduction of herniated contents, or gradual reduction during the first days of life by using a preformed or a customized silo (Schuster method). The survival rate in gastroschisis in developed countries is more than 90% [3], but still a very high morbidity and mortality is common in our country, because of the low rate of antenatal diagnosis, late closure of the abdominal wall, sepsis and abdominal compartment syndrome, with consequent morbidity and long period of hospitalization [4,5]. The aim of the present study is to detect the main factors of an unfavorable evolution in the case of gastroschisis and to find the best predictors of death, in order to subsequently improve the high rate of morbidity and mortality for this congenital malformation.

## 2. Materials and Methods

We conducted a retrospective cohort study of neonates with gastroschisis treated at our Pediatric Surgery Department. Prior to data extraction, approval was granted by the Ethics Committee of “Saint Mary” Emergency Children’s Hospital. Inclusion criteria for this analysis consisted of patients who were treated for gastroschisis from March 1990 to August 2020 in our tertiary care center. Eight patients with incomplete medical records or initially operated on elsewhere were excluded from the analysis. Patient data were extracted from the computerized database of the hospital and were statistically processed. We analyzed the demographic data, antenatal diagnosis, gestational age (GA), mode of delivery, APGAR score (**A**ppearance, **P**ulse, **G**rimace, **A**ctivity, and **R**espiration), gender, birth weight (BW), associated abnormalities, time to surgery, presence of compromised bowel (intestinal atresia or ischemic/necrotic bowels, meaning complex gastroschisis according to the Molik classification [6]), type of repair, post-operative complications (PC), biological investigations, sepsis, length of hospitalization (LH) and mortality. We will further define some terms used in the statistical analysis. Thus, birth weight is considered normal (NBW) if it is greater than or equal to 2500 g, and low (LBW) if it is less than 2500 g. Sepsis is defined as the presence of a positive blood culture or persistently abnormal clinical signs or inflammatory biomarkers and abnormal core temperature (>38.5 °C or <36 °C). Bronchopneumonia is defined by the presence of pathogens in the trachea or blood, as well as clinical and radiographic findings. Regarding the treatment of newborns with gastroschisis in our clinic, primary closure is attempted as a rule, based on the ease of reduction of abdominal contents, arterial oxygen saturation and the need for increased ventilatory pressures, objectively assessed by the pediatric surgeon and the anesthesiologist. If excess abdominal tension with compromised ventilation or disproportional abdominal cavity is noted, a delayed closure with placement of a silo is performed. The surgical approach did not change during the study period, but the conditions in the neonatal intensive care unit improved. Abdominal compartment syndrome (ACS) is a pathological increase of the intra-abdominal pressure above 15–20 mm Hg, with dysfunction of one or more organs.

### Statistics

We initially performed a descriptive statistical process, separately over the three decades. Then we investigated whether there is any link or reciprocal influence between the aforementioned statistical variables (antenatal diagnosis, GA, APGAR score, gender, type of birth, social environment, level of parental education, marital status of parents, toxic substances, surgical approach, anemia, thrombocytopenia, acute renal failure, heart abnormalities, enterocolitis, intestinal atresia, BW, ACS, sepsis and bronchopneumonia) and the risk of death of patients with gastroschisis. First, the chi-square test and the contingency tables (cross tabulation) were used in the analysis, the studied variables being of nominal type [7]. Subsequently, the logistic regression was used to determine the risk of death of patients with gastroschisis, a risk estimated based on independent variables previously validated by the chi-square test. The nominal factorial variables underlying the proposed model are: birth weight (LBW/NBW), abdominal compartment syndrome, sepsis and bronchopneumonia. The death-dependent variable is a nominal dichotomous variable that can take the values YES/NO. The regression equation used in the presented model is:$$ln\left(\frac{Prob\left(death\right)}{Prob\left(survive\right)}\right)=\beta_{0}+\beta_{1}\left(Weight\right)+\beta_{2}\left(ACS\right)+\beta_{3}\left(Sepsis\right)+\beta_{4}\left(Bronhopn.\right)$$
or
$$Odds=\frac{Prob\left(death\right)}{Prob\left(survive\right)}=e^{\beta_{0}}\times e^{\beta_{1}\left(Weight\right)}\times e^{\beta_{2}\left(ACS\right)}\times e^{\beta_{3}\left(Sepsis\right)}\times e^{\beta_{4}\left(Bronhopn.\right)}$$

All *p*-values were two-sided with values less than 0.05 considered statistically significant. SPSS statistical software (version 25) was used for this investigation.

A measure of goodness-of-fit often used to evaluate the fit of a logistic regression model is based on the simultaneous measure of sensitivity (true positive) and specificity (true negative) for all possible cutoff points. First, we calculate sensitivity and specificity pairs for each possible cutoff point and plot sensitivity on the y axis by (1-specificity) on the x axis using the ROC (Receiver Operating Characteristic) curve. The area under the ROC curve ranges from 0.5 and 1.0 with larger values indicative of better fit.

## 3. Results

One hundred and sixty-seven newborns with gastroschisis were admitted in our hospital during the 30 years and 159 were eligible for the study; we noticed a slight ascending trend of the appearance of the malformation during the last three decades, respectively 41/59/59 newborns (we have included 2020 in the last decade). Over the three decades analyzed, the rate of antenatal diagnosis and cesarean births increased significantly and the age at the time of surgery improved, as can be seen in Table 1. The survival rate increased constantly and the evolution of other parameters (GA, BW, LH) can also be observed in Table 1. The male/female ratio was 1.12. The rate of complex gastroschisis was constant, at approximately 25%. In 73% of patients it was possible to integrate the intestinal loops in the abdomen and close the abdominal wall per primam; the rest of the patients were treated by the Schuster method.

Reciprocal influence between statistical variables and death was statistically proved for only four independent variables: BW, ACS, sepsis and bronchopneumonia. It can be seen from the graphical representations (Table 2) that there is a direct and very obvious influence for these factors. Based on our study results, no association was observed between mode of delivery, GA, type of surgery or all the other factors and mortality among neonates with gastroschisis. The four independent variables included in the logistic regression model are presented in Table 2.

According to omnibus tests of model coefficients, all of the variables entered in the equation for our model have a significant effect (chi-square = 68,200; df = 4; Sig. = 0.000).

In order to estimate the coefficient of determination as a model summary, we used Nagelkerke’s R-square which shows us that all four independent variables contribute 47.6% to the risk of death.

Goodness-of-fit statistics are very useful to determine whether the model adequately describes the data. The Hosmer–Lemeshow statistic indicates a poor fit if the significance value is less than 0.05. In our regression equation, we observed that the model adequately fits the data (chi-square = 11,410; df = 7; Sig. = 0.122).

The classification table (Table 3) compares the predicted values for the dependent variable, based on the regression model, with the actual observed values in the data. In our regression model, the four independent variables considered can predict which value of the dependent one (death) is observed in the dataset 81.1% of the time (85.0% for YES and 74.6% for NO).

In the final table, Variables in the equation (Table 4), we can see that each independent variable considered has a significant impact on the predicted variable (Sig. < 0.05). The estimated levels of the regression coefficients *β_i_* are marked with B and EXP (B) represents odds ratio for each factorial variable, which is $e^{\beta_{i}}$.

Therefore, we shall have the logistic regression equation:$$\frac{\mathrm{Prob}\left(\mathrm{death}\right)}{\mathrm{Prob}\left(\mathrm{survive}\right)}=\left(0.107\right)\times \left(4.908\right)\;\mathrm{Weight}\times \left(7.782\right)\mathrm{ACS}\times \left(2.997\right)\;\mathrm{Sepsis}\;\times \left(7.883\right)\;\mathrm{Bronhopneumonia}$$

If the BW is below normal then we have an increased risk (4908 times) of evolution to death. Similarly, as seen from the logistic regression equation, the risk of death is 7782 times higher in the case of developing ACS, about three times in the case of sepsis, and 7883 times in the case of bronchopneumonia. All 4 independent variables contribute 47.6% to the risk of death.

In our analysis by logistic regression method, we save the composed probabilistic value of the 4 variables (predicted probability PRE_1) from the proposed model (sepsis, bronchopneumonia, birth weight, abdominal compartment syndrome) and then we will use this value as a test in the graph of the ROC curve together with the dependent variable (state variable) death.

The area under the curve (Figure 1) is 0.850 with 95% confidence interval (0.784, 0.916). Also, the area under the curve is significantly different from 0.5 since *p*-value is 0.000 meaning that the diagnostic accuracy of our previously presented logistic regression model is very good for predicting the contribution of the four independent variables to the risk of death (Table 5).

## 4. Discussion

Gastroschisis is a congenital abdominal wall defect that has presented an increasing occurrence in past decades at great cost for the health system [5]. Although the survival rate has been increasing in our country, the morbidity continues to be high and the length of hospitalization extended. A few independent risk factors for mortality among gastroschisis patients have been reported previously in the literature. Special efforts are being made throughout the world to detect those determinants of increased morbidity, in order to eliminate or improve them. Raymond et al. found that the presence of complex gastroschisis, preterm delivery, and very low birth weight were associated with worse clinical outcomes including increased sepsis, short bowel syndrome, parenteral nutrition days, and hospital length of stay [3], our results being somewhat similar. Sepsis, birth weight <2500 g, an additional congenital anomaly, and gestational age <34 weeks were also identified [8,9,10]. The risk of death is increased in newborns with gastroschisis who have liver herniation, pulmonary hypoplasia, abdominal compartment syndrome, relaparotomy for perforation or necrosis, or central line-associated sepsis. Special care should be taken to minimize the risk of central line sepsis in the clinical setting [11].

Elective preterm delivery by cesarean section appears favorable with respect to intestinal injury, feeding and sepsis [12]. Although this type of birth could not be demonstrated to be beneficial in the long-term evolution of patients, there appears to be a direct correlation between the rate of antenatal diagnosis and cesarean delivery in our study. Nevertheless, while death was not associated with mode of delivery either in our study or in other similar studies, we cannot rule out the possibility that that there are clinically beneficial aspects of a cesarean delivery that might be noticeable for less severe outcomes [2]. In any case, the significant increase (*p* = 0.000) of the antenatal diagnosis rate led in our country to the significant increase of the number of births by cesarean section and implicitly to the decrease of GA, as observed in Table 1. The birth rate by cesarean section in the last decade in our country is similar to that reported by other recent studies conducted in developed countries, i.e., 50 versus 55% [13].

There has been a lot of controversy about GA at delivery. Some studies suggest that elective earlier delivery anticipation may reduce intestinal damage and enhance neonatal outcome [13]; this could not be demonstrated in our study. We must not lose sight of the fact that the decrease of GA implicitly leads to the decrease of BW, the latter factor being directly correlated with an unfavorable evolution and increasing by 4.90 times the risk of death in our study.

Currently, the most important determinant of gastroschisis outcomes in high resource settings is whether the condition is associated with intestinal complications, such as atresia, necrosis, perforation, or volvulus. Although some report that complex cases occur in up to one third of pregnancies affected by gastroschisis [14,15], the rate of complex gastroschisis in our study was approximately 25%. This form of the anomaly accounts for most of the mortality and a disproportionate burden of the morbidity from gastroschisis [16]. Due to the discrepancy between the low rate of complex cases and the high mortality rate in our study compared to other studies, we think that perhaps complex gastroschisis was underdiagnosed in our case.

The technique for surgical closure depends on the degree of intestinal inflammation or other intestinal abnormalities, the size of the defect, and the general conditions of the newborn. The classical approach is to perform primary closure in the first hours of life, with the newborn under general anesthesia, in an operating room; this was done in 73% of our cases. Silo placement and delayed closure is indicated in cases with viscero-abdominal disproportion due to the risk of compartment syndrome. ACS is directly correlated with an unfavorable evolution, increasing the risk of death 7.78 times in our study and we must avoid this complication in the future through stricter intra and postoperative monitoring of abdominal pressure. 19 of our patients (12%) developed ACS, a complication that could have been avoided by adopting the Schuster method instead of primary closure for patients with viscero-abdominal disproportion. High intra-abdominal pressure can lead to vascular compromise and bowel ischemia resulting in complications. The risk of ACS has been reported to be significantly more common among patients undergoing direct closure of congenital abdominal wall defect [17]. A recent study demonstrates largely equivalent outcomes between patients who underwent immediate closure and those who had a silo less than 5 days and in fact reduced ventral hernias among those patients [18]; the Schuster method should not be avoided for fear of complications such as infections or the subsequent onset of a ventral hernia.

Although the value of *p* did not reach the threshold of statistical significance (*p* = 0.083 between the second and third decade), the constant decrease in age at the time of surgery was certainly another positive factor in significantly increasing the survival rate in our group. The constant and significant increase of the antenatal diagnosis rate also contributed to this positive aspect.

As in other studies [15,19], sepsis and bronchopneumonia were major causes of increased mortality in our case. The risk of death is about three times higher in the case of sepsis and 7.883 times higher in the case of bronchopneumonia. Antenatal diagnosis, bringing the newborn as soon as possible to the neonatal intensive care unit, closing the abdominal defect in the best aseptic conditions, avoiding the ACS and early antibiotic therapy are the conditions to prevent wound infections and early sepsis. Late-onset neonatal sepsis is caused by microorganisms acquired from the environment after childbirth. Recent advances in the approach to late-onset neonatal sepsis have resulted in a significant increase in survival, even in the face of prolonged hospitalizations, mechanical ventilation, use of invasive procedures and devices (intravascular catheters and endotracheal cannulas), which are predisposing factors for this condition [19].

According to the ROC curve, the diagnostic accuracy of our presented logistic regression model was very good for predicting the contribution of the four independent variables to the risk of death. All four independent variables contributed 47.6% to the risk of death in our study.

### 4.1. Limitations

There are limitations intrinsic to this study as the review was retrospectively performed and it is a single-center study. This study included cases from 30 years, which means many medical knowledge and therapeutic concepts might have been different in more recent decades. In a retrospective review, this issue will lead to bias.

### 4.2. Strenghts of the Study

Strengths of the study: the results of this study are drawn from the largest population-based study in our country, our hospital serving the entire northeastern region of Romania, which in mid-2020 accounted for 18% of Romania’s population, numbering approximately 4 million people. However, we cannot generalize the results obtained in our region to the entire country.

## 5. Conclusions

In the past 30 years in our country we have seen transformational improvements in outcomes of gastroschisis due to advances in neonatal intensive care and enhanced integration between the disciplines of maternal fetal medicine, neonatology and pediatric surgery. However, some factors such as sepsis, bronchopneumonia, abdominal compartment syndrome and low birth weight may still be ameliorated for better outcomes in these patients.

## Figures and Tables

**Figure 1 jpm-11-01168-f001:**
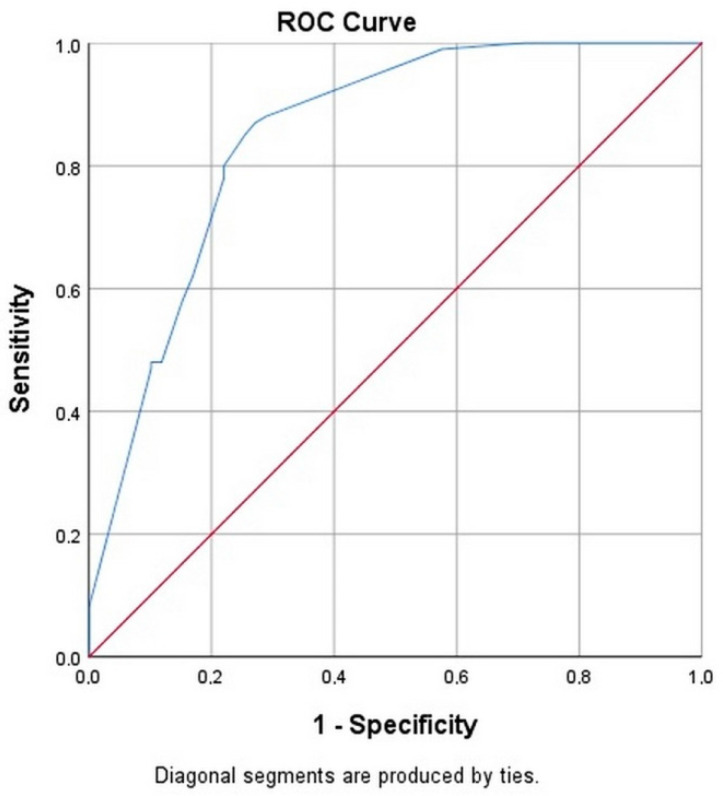
Composite ROC curve for the logistic regression model.

**Table 1 jpm-11-01168-t001:** Descriptive statistics of the analyzed patients.

Years	1990–1999	2000–2009	2010–2020	Overall
Frequency of occurrence	41	59	59	159
Mothers age (between 13 and 40 years old)	21.09	20.55	21.20	20.94
The rate of antenatal diagnosis	0%	12%	53%	23.2%
*p*-value			0.000	
The rate of cesarean births	10%	20%	50%	28.3%
*p*-value			0.0005	
Gestational age (weeks)	36.7	36.6	35.9	36.4
*p*-value		0.571	0.070	
Birth weight (between 1000 and 3800 g)	2320.97	2343.72	2272.03	2311.25
*p*-value		0.598	0.235	
The age at the time of surgery (hours)	7.47	7.25	6.10	6.87
*p*-value		0.405	0.083	
Age at death (days)	7.23	13.71	28.27	15.85
Length of hospitalization for survivals (days)	28.82	36.47	41.68	36.66
The survival rate	26.8%	28.8%	52.5%	37.1%
*p*-value		0.587	0.004	

*p*-value was calculated for the differences between the first and the second decade, and between the second and the third decade respectively.

**Table 2 jpm-11-01168-t002:** Categorical variable codings and chi-square tests for statistical variables considered individually and the risk of death.

Independent Variables		Frequency	Parameter Coding	Pearson Chi-Square (Each Independent Variable vs. Death)		
			(1)	Value	df	Asymptotic Significance (2-Sided)
Birth weight	LBW	106	1	15.577 a	1	0.000
	NBW	53	0			
Sepsis	Yes	88	1	20.329 a	1	0.000
	No	71	0			
Bronchopneumonia	Yes	88	1	42.122 a	1	0.000
	No	71	0			
Abdominal Compartment Syndrome	Yes	19	1	4.202 a	1	0.040
	No	140	0			

a. 0 cells (0.0%) have expected count less than 5.

**Table 3 jpm-11-01168-t003:** Classification table.

	Observed		Predicted		
			DEATH		Percentage Correct
			NO	YES	
Step 1	DEATH	NO	44	15	74.6
		YES	15	85	85.0
	Overall Percentage				81.1

The cut value is 0.500.

**Table 4 jpm-11-01168-t004:** Variables in the equation.

		B	S.E.	Wald	df	Sig.	Exp(B)
Step 1	Abdominal compartment syndrome (1)	2.052	0.760	7.292	1	0.007	7.782
	Sepsis (1)	1.097	0.482	5.176	1	0.023	2.997
	Bronchopneumonia (1)	2.065	0.472	19.162	1	0.000	7.883
	Birth weight (1)	1.591	0.459	12.010	1	0.001	4.908
	Constant	−2.237	0.495	20.412	1	0.000	0.107

**Table 5 jpm-11-01168-t005:** Test result variable(s): predicted probability.

Area	Std. Error ^a^	Asymptotic Sig. ^b^	Asymptotic 95% Confidence Interval	
			Lower Bound	Upper Bound
0.850	0.034	0.000	0.784	0.916

The test result variable(s): predicted probability has at least one tie between the positive actual state group and the negative actual state group. Statistics may be biased. ^a^ Under the nonparametric assumption. ^b^ Null hypothesis: true area = 0.5.

## Data Availability

The data that support the findings of this study are available from the corresponding author upon reasonable request.

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
