# Peer review of "Statistical Analysis of the Main Risk Factors of an Unfavorable Evolution in Gastroschisis"

_jpm, 2021, doi:10.3390/jpm11111168_

Round 1

Reviewer 1 Report

This is a very interesting article on newly found risk factors of unfavorable outcomes (death) of gastroschisis. I must congratulate the authors on presenting a formula for predicting the death of an infant due to the summarization of risk factors. I think this is the most important finding of this work and it is of the utmost clinical relevance. However, in my opinion, it is not emphasized enough. 

The introduction is well written.

Materials and methods are satisfying however I have some minor issues to be corrected:

1. The four most important variables, especially when Boolean, should be defined more precisely. E.g. LBW is defined as over or under 2500g and the others should be defined the same. What exactly is ACS? Do we measure intraabdominal pressure or it is a clinical entity with organ failure, what is it by definition in a newborn (we need precise information). The same applies to sepsis - how is it defined in your work? Bronchopneumonia as well.

2. Minor issue is GA - acronym used in the discussion, table 1, etc - it should be defined when firstly stated - presumably in Line 62 in Methods. Also, these acronyms should be added to tables - GA, BW so tables can be self-standing or it should be stated in table1 as Length of hospitalization (LH) in full wording. 

2a. Consider adding a flowchart of included/excluded patients according to reasons.

The result section should be in one way shortened, and in other broadened  technically what I mean is the following:

3. Table 1 - there should be a p-value of the test indicating the difference between the indices according to stated decades. Since the authors observe GA decrease (not sure if statistically relevant), decrease in BW etc.

4. Tables 2 and 3 could be merged and presumably the risk factors described in the table (as I proposed in the methods section).

5. Tables 4, 5, and 6 represent the results of the regression model and could be just stated that the overall fit of the model is good by the HL test, and the variance of the negative outcome is explained from C&S 0.349 to Nagelkerke 0.476. The current data can be stated fully in a supplementary file.

6. What I miss the most are ROC curves (or a composite ROC curve) stating the metric of the prediction along with table 7.

Discussion section is very well written although wordy, but if no word limit I would leave it intact.

7. A lot of the discussion about own results is according to table 1 with no actual statistical test to validate the observed differences (e.g. in lines 179 through 174 there is a statement of increase of antenatal diagnosis...). As I stated in point #3.

Author Response

Response to Reviewer 1 Comments

Dear Reviewer,

Thank you very much for evaluating our manuscript. Your recommendations and comments have helped us greatly improve our manuscript. Here we provide the requested corrections and address the comments. The changes we have made in the manuscript are highlighted in red.

Point 1: The four most important variables, especially when Boolean, should be defined more precisely. E.g. LBW is defined as over or under 2500g and the others should be defined the same. What exactly is ACS? Do we measure intraabdominal pressure or it is a clinical entity with organ failure, what is it by definition in a newborn (we need precise information). The same applies to sepsis - how is it defined in your work? Bronchopneumonia as well.

 Response 1: In the Methods section we defined the terms used in statistical analysis. Thus, birth weight is considered normal (NBW) if it is greater than or equal to 2500 grams, and low (LBW) if it is less than 2500 grams. Sepsis is defined as the presence of a positive blood culture or persistently abnormal clinical signs or inflammatory biomarkers and abnormal core temperature (>38.5° or <36°C). Bronchopneumonia is defined by the presence of pathogens in the trachea or blood, as well as clinical and radiographic findings. Regarding the treatment of newborns with gastroschisis in our clinic, primary closure is attempted as a rule, based on the ease of reduction of abdominal contents, arterial oxygen saturation and the need for increased ventilatory pressures, objectively assessed by the pediatric surgeon and the anesthesiologist. If excess abdominal tension with compromised ventilation or disproportional abdominal cavity is noted, a delayed closure with placement of a silo is performed. Abdominal compartment syndrome (ACS) is a pathological increase of the intra-abdominal pressure above 15-20 mm Hg, with dysfunction of one or more organs.

Point 2: Minor issue is GA - acronym used in the discussion, table 1, etc - it should be defined when firstly stated - presumably in Line 62 in Methods. Also, these acronyms should be added to tables - GA, BW so tables can be self-standing or it should be stated in table1 as Length of hospitalization (LH) in full wording.

Response 2: In the Methods section we defined the acronym GA (gestational age). In table 1 we replaced the acronyms with the full wording.

Point 2a: Consider adding a flowchart of included/excluded patients according to reasons.

Response 2a: In the Methods section we defined the selection criteria:Inclusion criteria for this analysis consisted of patients who were treated for gastroschisis from March 1990 to August 2020 in our tertiary care centre. Eight patients with incomplete medical records or initially operated elsewhere were excluded from the analysis”. We avoided adding a flowchart in order not to repeat the data presented in the text and not to load the article unnecessarily.

Point 3: The result section should be in one way shortened, and in other broadened  technically what I mean is the following: Table 1 - there should be a p-value of the test indicating the difference between the indices according to stated decades. Since the authors observe GA decrease (not sure if statistically relevant), decrease in BW etc.

 Response 3: We calculated the value of p for some of the variables presented in table 1 and we added the comments about them in the results chapter as well as in the discussions.

 Point 4: Tables 2 and 3 could be merged and presumably the risk factors described in the table (as I proposed in the methods section).

 Response 4: Table 2 and 3 were merged and the risk factors described.

 Point 5: Tables 4, 5, and 6 represent the results of the regression model and could be just stated that the overall fit of the model is good by the HL test, and the variance of the negative outcome is explained from C&S 0.349 to Nagelkerke 0.476. The current data can be stated fully in a supplementary file.

 Response 5: Tables 4, 5 and 6 were eliminated and the data were included in the text.

 Point 6: What I miss the most are ROC curves (or a composite ROC curve) stating the metric of the prediction along with table 7.

 Response 6: We included a composite ROC Curve in our analysis and validated the logistic regression model. Thank you for the good advice.

 Point 7: Discussion section is very well written although wordy, but if no word limit I would leave it intact. A lot of the discussion about own results is according to table 1 with no actual statistical test to validate the observed differences (e.g. in lines 169 through 174 there is a statement of increase of antenatal diagnosis...). As I stated in point #3.

 Response 7: We calculated the p-value of variables presented in table 1 and we added the comments about it in the results chapter as well as in the discussions.

 Thank you again for reviewing our manuscript,

Elena Țarcă, MD, PhD

Reviewer 2 Report

Dear author

Congratulation to complete this excellent task and your finding makes impact to predict the prognosis of gastroschisis in past 30 years. 

I have some comments about this study.

First, there are many tables. In fact, you can summarized them into one table which can simplified the audience to read your article smoothly. Table 2 & 3 can be one, and table 4-7 can summarized into one .

Second, this study included cases from 30 years, which means many medical knowledge and therapeutic concepts might be different in these decades. In a retrospective review, this issue will lead bias. Please comment this . And if possible, I suggest you can describe the difference of management of these children in past decades in Method section.

Third, the survival rate is obvious elevated in past ten years comparing with last two decades. You can analyze and present the favorable factors to improve survival. 

Fourth, the structure of this manuscript is loose and some descriptions or details are not necessary (ex. YES/NO ). They will distract the attention of the readers. Please concise this manuscript.

Fifth, it is appreciated if the discussion focus on the findings of this study. Some finding of this study is not deeply discussed. please comment it.

Thank you for the honor to review your manuscript.

Best,

Author Response

Response to Reviewer 2 Comments

Dear Reviewer,

Thank you very much for evaluating our manuscript. Your recommendations and comments have helped us greatly improve our manuscript. Here we provide the requested corrections and address the comments. The changes we have made in the manuscript are highlighted in red.

Point 1: First, there are many tables. In fact, you can summarized them into one table which can simplified the audience to read your article smoothly. Table 2 & 3 can be one, and table 4-7 can summarized into one .

Response 1: Table 2 and 3 were merged; Tables 4, 5 and 6 were eliminated and the data were included in the text.

Point 2: Second, this study included cases from 30 years, which means many medical knowledge and therapeutic concepts might be different in these decades. In a retrospective review, this issue will lead bias. Please comment this . And if possible, I suggest you can describe the difference of management of these children in past decades in Method section.

Response 2: Thank you for this observation; we added at the end a sentence in which we drew attention to this aspect. In the Methods section we described the main therapeutic aspects applied.

Point 3: Third, the survival rate is obvious elevated in past ten years comparing with last two decades. You can analyze and present the favorable factors to improve survival.

Response 3: The survival rate has significantly improved in the last decade studied compared to the others; we statistically analyzed these data in table 1 and we tried to present the factors that favored this aspect in the Discussion section.

Point 4: Fourth, the structure of this manuscript is loose and some descriptions or details are not necessary (ex. YES/NO ). They will distract the attention of the readers. Please concise this manuscript.

Response 4:  We eleminated some tables and the unnecessary words.

Point 5: Fifth, it is appreciated if the discussion focus on the findings of this study. Some finding of this study is not deeply discussed. please comment it.

Response 5: We tried to emphasize, in the Discussions section, the most important results of our study. We have added a few sentences in this regard. For validating our logistic regression model, we also added a composite ROC Curve.

Thank you again for reviewing our manuscript,

Elena Țarcă, MD, PhD

Round 2

Reviewer 1 Report

I must congratulate the authors again on an excellent presentation of the results. I am very grateful to have reviewed this work. Also, I am very pleased to see that all of my suggestions have been acknowledged, accepted, and added to the work. 

Just two small technical corrections - be sure to add the name of the statistical test used in the legend of Table 1 (with the description of the p-value) and in line 168 rephrase/depersonalize "...we can see that...".

Reviewer 2 Report

Congratulation again.

All of my concerns were fulfilled. Your effort to make this interesting article more valuable to read.

Best,